# Resolución del problema de conjunto dominante de influencia positiva mínima en redes sociales mediante metaheurísticas

**Iván Penedo**
Dpto. de Informática y Estadística
Universidad Rey Juan Carlos
Móstoles, España
ivan.penedo@urjc.es

**Isaac Lozano-Osorio**
Dpto. de Informática y Estadística
Universidad Rey Juan Carlos
Móstoles, España
isaac.lozano@urjc.es

**Jesús Sánchez-Oro**
Dpto. de Informática y Estadística
Universidad Rey Juan Carlos
Móstoles, España
jesus.sanchezoro@urjc.es

**Oscar Cordón**
Dpto. de Ciencias de la Computación e IA
Universidad de Granada
Granada, España
ocordon@decsai.ugr.es

## Abstract

El auge de internet y las redes sociales han propuesto nuevos retos para intentar estudiar el comportamiento de las personas en ellas. Normalmente, cada persona tiene gustos similares a los de un pequeño grupo de personas, aquellos que son más afines o en los que más confían. Por este motivo, se han diseñado nuevas técnicas de marketing viral que pretenden propagar la información sobre el producto o servicio que quieran anunciar de manera efectiva. De esta motivación, se definen los problemas de maximización/minimización de influencia social y conjuntos de dominancia. En concreto, el problema de conjuntos dominantes de influencia positiva mínima (del inglés Minimum Positive Influence Dominating Set, MPIDS) consiste en obtener un conjunto de dominancia de cardinalidad mínima que permita influir en toda una red social. Como restricción, el problema define que para que un nodo se vea influenciado, al menos la mitad de sus vecinos deben encontrarse en el conjunto de dominancia. Teniendo en cuenta que el MPIDS es un problema $\mathcal{NP}$-difícil donde aproximaciones exactas no son posibles debido al tamaño de las redes sociales. En este trabajo se propone el uso de *Greedy Randomized Adaptive Search Procedure* (GRASP). Esta reconocida metaheurística se compone de dos fases: una fase constructiva y una fase de mejora. La fase constructiva diseñada se compara con un enfoque completamente voraz y otro completamente aleatorio. Por otro lado, la fase de mejora se lleva a cabo en dos etapas. En la primera, se aplica un procedimiento de eliminación de nodos redundantes dentro del conjunto solución. Posteriormente, se propone una estrategia innovadora de búsqueda local, basada en la generación de agujeros, que consiste en eliminar la $\delta$-vecindad de un nodo para facilitar una reconstrucción inteligente de la solución. Los resultados obtenidos muestran una desviación de un 5.54 % frente al estado del arte sobre un conjunto de 196 instancias de tamaños variables.

## 1. Introducción

Los problemas de maximización de influencia social (*Social Influence Maximization*, SIM) han sido ampliamente estudiados en el contexto del marketing viral, donde los consumidores influyen

XVI Congreso Español de Metaheurísticas, Algoritmos Evolutivos y Bioinspirados (maeb 2025).

secuencialmente sobre sus relaciones sociales para la compra de un producto [4]. Algunos trabajos relacionados muestran que las interacciones entre usuarios en una red social pueden ser utilizadas para prevenir la propagación de enfermedades [9], realizar campañas de marketing viral [12], mostrar recomendaciones en software de aprendizaje en línea [16], estudiar relaciones sociales [8] o prevenir el abuso de tabaco u otras sustancias [22]. Para estos problemas, se busca identificar aquellos usuarios que aceleran o reducen la difusión de influencia en la red [14, 19].

El problema de conjuntos dominantes de influencia positiva mínima, o en inglés *Minimum Positive Influence Dominating Set (MPIDS)* busca encontrar el conjunto de usuarios de tamaño mínimo para conseguir influenciar a toda la red social. Este problema se definió formalmente en [23] donde se demostró que es un problema $\mathcal{NP}$-difícil para grafos generales. Asimismo, en el ámbito de estos problemas, una red social se modela formalmente como un grafo no dirigido $G = (V, A)$. Cada arista $(u, v) \in A$ tiene dos extremos $u$ y $v$, indicando que estos usuarios están conectados, por tanto, un usuario tiene relación con otro y puede influenciarle en la red social representada. Además, se define $N(v)$ como el conjunto de vértices adyacentes a $v$, i.e., $\{u \in V : (u, v) \in A\}$.

Dado un grafo no dirigido $G = (V, A)$, un conjunto dominante de influencia positiva (PIDS) trata de buscar un conjunto de vértices $D \subseteq V$ tal que al menos la mitad de los vecinos de cualquier vértice se encuentre en $D$, i.e., $|D \cap N(v)| \geq |N(v)|/2 \ \forall v \in V$. Concretamente, el objetivo del problema MPIDS es el de obtener un conjunto dominante de influencia positiva con la mínima cardinalidad. Lin et al. [11] definieron el siguiente modelo de programación entera lineal 1.

$$
\begin{aligned}
\textbf{minimizar} \quad & \sum_{i=1}^{|V|} x_i \\
\textbf{sujeto a} \quad & \sum_{v_j \in N(v_i)} x_j \geq \left\lceil \frac{|N(v_i)|}{2} \right\rceil \quad \forall i \in V \\
& x_i \in \{0,1\} \quad \forall i \in V
\end{aligned}
\tag{1}
$$

La variable binaria $x_i \in \{0, 1\}$ es asignada a cada vértice $i \in V$, donde $x_i = 1$ indica que el vértice $i$ es seleccionado, mientras que $x_i = 0$ implica lo contrario. Por otra parte, la restricción principal obliga a que cualquier solución factible tenga seleccionados los vértices necesarios para que todos los vértices $v_i \in V$ tengan al menos la mitad de sus vecinos seleccionados en $D$.

El principal inconveniente es el tamaño que hoy en día tienen las redes sociales, que hacen que los modelos matemáticos no puedan en algunos casos ni siquiera proporcionar una solución factible. Por ello, existen diferentes aproximaciones voraces o heurísticas en la literatura.

En primer lugar, Wang et al. [23] plantearon un algoritmo voraz que en cada iteración seleccionaba el vértice que podría influenciar a un mayor número de vértices con una complejidad algorítmica de $O(|V|^3)$. Por otra parte, Raei et al.[18] definieron el parámetro *cover-degree* para priorizar ciertos vértices para el criterio voraz reduciendo la complejidad en tiempo a $O(|V|^2)$. Posteriormente, el algoritmo *Fast Greedy Algorithm* (FGA) presentado por Pan [17] seguía la misma estrategia voraz pero propagándose por la vecindad del último nodo dominado y priorizando la dominancia sobre los nodos que no satisfacen la restricción, reduciendo la complejidad a $O(|V| \log |V| + |E|)$. El algoritmo *Improved Greedy Algorithm* IGA-PIDS [3] está basado en FGA [17] y utilizaba el parámetro *need-degree* además de *cover-degree* para priorizar nodos, además de realizar una poda del grafo al inicio para obtener aquellos nodos que deben estar dominados siempre y una última criba para no dominar los nodos redundantes en el conjunto de dominancia al final de la ejecución del algoritmo. Esta última propuesta, aun manteniendo una complejidad de tiempo de $O(|V| \log |V| + |E|)$, obtiene soluciones de mayor calidad que FGA.

Por otra parte, se pueden encontrar diferentes algoritmos metaheurísticos en la literatura. Uno de los primeros fue el algoritmo memético ILPMA de Lin et al. [11], que presentaba la formulación matemática del problema, además de proponer una búsqueda tabú para realizar optimizaciones locales en el conjunto de dominancia. Más adelante, el algoritmo *Iterated Carousel Greedy* (ICG) sustituyó la búsqueda tabú por un proceso de iterativo de destrucción y reconstrucción del conjunto de dominancia. La primera variante de un algoritmo CMSA de este tipo, propuesta por Akbay y Blum [1], generaba varias soluciones para posteriormente mezclarlas y realizar varias iteraciones utilizando estructuras adaptables auxiliares para variar la generación de conjuntos de dominancia. Estas estructuras fueron sustituidas por una variable $n_a$ que denotaba el número de soluciones

generadas en cada iteración, el cual se incrementaba al encontrar una solución con el mismo valor de la función objetivo y se restablecía al encontrar una mejor solución, en la variante auto-adaptativa del algoritmo [2]. Finalmente, el algoritmo FastPIDS de Sun et al. [21] propone varias reglas de reducción para la generación de conjuntos de dominancia, así como reutilizar la función *need-degree* de la literatura. Asimismo, el algoritmo combina una construcción voraz utilizando un criterio híbrido con una búsqueda local basada en el intercambio de vértices y el mecanismo de *two-level satisfaction judgment* (TLSJ). Este mecanismo utiliza dos proposiciones para conocer qué conjuntos parciales de dominancia son prometedores y cuáles no para añadir nuevos vértices a este conjunto utilizando la heurística híbrida. El algoritmo FastPIDS es actualmente el de mejor rendimiento según la literatura, por lo que lo usaremos para comparar frente a nuestra propuesta.

Este trabajo presenta una aproximación heurística para aportar soluciones de calidad al problema de MPIDS con un tiempo de cómputo reducido. Se ha diseñado un algoritmo *Greedy Randomized Adaptive Search Procedure (GRASP)*, proponiendo una novedosa mejora basada en crear agujeros en la solución para después aplicar una reconstrucción inteligente del conjunto de dominancia.

El artículo se organiza como sigue. El diseño de la propuesta se realiza en la sección 2, que describe la etapa constructiva y la fase de mejora utilizando el proceso de búsqueda local. La sección 3 presenta los resultados obtenidos en los diferentes experimentos realizados. Estos se compararán con los resultados del mejor algoritmo existente para el problema de MPIDS para realizar una comparativa justa. Finalmente, en la sección 4 se mencionan las conclusiones que se han alcanzado con el desarrollo de este trabajo y los trabajos futuros.

## 2. GRASP

La metaheurística GRASP fue desarrollada [5] y formalizada [6] por Feo et al. como una técnica de optimización multi-arranque dividida en dos etapas. En la primera se genera una solución voraz y aleatoria parametrizada con $\alpha$, parámetro que define cómo de voraz o aleatoria es la construcción. Después se aplica un método de mejora que alcance un óptimo local en una vecindad definida. Estas fases se repiten hasta alcanzar un criterio de parada, devolviendo la mejor solución obtenida.

En el Algoritmo 1 se puede observar el pseudocódigo de la metodología planteada. Es necesario disponer del grafo $G$ y de los valores de los parámetros $\alpha$ y $\delta$, y del criterio de parada que, en este trabajo, se define como un tiempo máximo de ejecución. El algoritmo comienza construyendo una solución inicial con todos los vértices $V$ del grafo (línea 1), tratándose de una solución trivial al problema. A continuación, se itera hasta alcanzar el criterio de parada establecido (línea 2). En cada una de las iteraciones, se construye una solución inicial utilizando el parámetro $\alpha$ (línea 3), para después eliminar aquellos nodos redundantes en el conjunto de dominancia (línea 4). A la solución obtenida se le aplica la búsqueda local (línea 5) para alcanzar un óptimo local en la vecindad definida. En caso de mejora, se actualiza la mejor solución encontrada hasta el momento, $D_b$ (líneas 6-8). Finalmente, se devuelve la mejor solución global obtenida, el conjunto de dominancia $D_b$ (línea 10).

---

**Algoritmo 1** *GRASP* $(G = (V, A), \alpha, \delta, t_{\text{máx}}) \rightarrow D$

1: $D_b \leftarrow V$
2: **while** *tiempo* $<= t_{\text{máx}}$ **do**
3:     $D \leftarrow Construir(G, \alpha)$
4:     $D \leftarrow EliminarRedundantes(D)$
5:     $D \leftarrow Mejorar(D, \delta)$
6:     **if** $|D| < |D_b|$ **then**
7:         $D_b \leftarrow D$
8:     **end if**
9: **end while**
10: **return** $D_b$

---

Dado que el proceso constructivo puede provocar que la solución propuesta contenga nodos que no sean necesarios para satisfacer las restricciones del problema, se ha añadido un proceso de refinamiento. Este proceso recorre los nodos añadidos al conjunto de dominancia para comprobar cuáles pueden ser eliminados manteniendo una solución factible. Para que un nodo $d \in D$ se considere redundante y, por lo tanto, se elimine de la solución, debe cumplirse la siguiente restricción:

$$|N(v) \cap D| > \left\lceil \frac{N(v)}{2} \right\rceil \forall v \in N(d)$$

Por tanto, el método recorrerá todos los nodos pertenecientes a la solución y eliminará aquellos que cumplan la restricción anterior.

## 2.1. Fase constructiva

La fase constructiva busca obtener una solución inicial partiendo de una solución vacía y ayudándose de un criterio voraz para seleccionar los nodos candidatos que añadir al conjunto de dominancia.

Es importante destacar que se han aplicado algunas de las reducciones planteadas por Sun et al. [21], que permiten determinar si ciertos nodos deben pertenecer o no a la solución por diferentes características de los nodos que se muestran a continuación. La primera reducción muestra que, si un nodo hoja tiene un único nodo padre, para poder obtener una solución factible es necesario que el padre siempre esté en el conjunto $D$. Por otro lado, en caso de tener una estructura de nodos formando un triángulo entre las relaciones, será necesario que la solución $D$ contenga dos de los tres nodos.

Una vez se han detallado las reducciones, se presenta el criterio voraz seguido en este trabajo. Este criterio se basa en el grado $|N(v)|$ de cada uno de los nodos $v \in V$. En la fase constructiva es necesario evaluar la contribución de cada uno de sus nodos con un criterio voraz. Los problemas de influencia en redes sociales utilizan conjuntos de datos con millones de usuarios. Esto provoca que analizar el criterio voraz para cada uno de los nodos sea un proceso computacionalmente costoso. Con el objetivo de aumentar la eficiencia del método y la escalabilidad, en este trabajo se alternan las fases aleatoria y voraz dentro de la fase constructiva de GRASP. Este método constructivo se denominará *Randomized Greedy Algorithm* (RGA). En concreto, el RGA utilizará el parámetro $\alpha$ para determinar un subconjunto de nodos, seleccionando entre ellos el candidato de mayor grado.

En un primer lugar, generamos una la lista de vértices *NF* formada por aquellos vértices que aún no cumplen la función de factibilidad tal que

$$NF = \left\{ v \in V : |N(v) \cap D| < \left\lceil \frac{|N(v)|}{2} \right\rceil \right\}$$

Es importante destacar que, para que una solución sea factible, es necesario lograr que $NF = \emptyset$, lo que define el criterio de parada del método constructivo.

A continuación, se elige un elemento aleatorio $v$ de *NF*, y se genera la lista de candidatos *LC* que pueden entrar a formar parte de la solución, formada por $N(v) \setminus D$, siendo $D$ la solución bajo construcción.

Una vez se dispone de la *LC*, se crea la lista de candidatos restringida (*LCR*). Mientras que en el esquema tradicional dicha lista incluye los candidatos más prometedores, en la propuesta de este trabajo se conforma con un porcentaje $\alpha$ de candidatos obtenidos al azar de la *LC*, que son los únicos evaluados a posteriori. El motivo de esta modificación es que, dado el tamaño de las redes sociales, la evaluación de todos los candidatos de la *LC* requiere un elevado tiempo de cómputo. El siguiente vértice a añadir a la solución será aquel que presente un mayor grado, i.e., $\max_{c \in LCR} |N(c)|$. Se continuarán añadiendo candidatos hasta que el nodo original $v$ cumpla las restricciones de factibilidad, momento en el que se eliminará de *NF* y se elegirá un nuevo candidato.

Como es tradicional en GRASP, el parámetro $\alpha$ controla el grado de aleatoriedad del método. En concreto, un valor de $\alpha = 1$ denotaría un criterio totalmente voraz (ya que evaluaría todos los candidatos de la *LCR*), mientras que un valor $\alpha = 0$ definiría uno totalmente aleatorio (ya que solo evaluaría un candidato al azar).

## 2.2. Fase de mejora

La propuesta novedosa en este trabajo consiste en una búsqueda local basada en la generación de agujeros (véase Algoritmo 2). Este método se propone con el objetivo de eliminar los nodos de una determinada región de la solución. Tras llevar a cabo la eliminación, se reconstruirá la solución de manera inteligente para obtener mejores soluciones.

El algoritmo itera hasta que no encuentre ninguna mejora en todo el conjunto solución o alcance el límite de tiempo de ejecución (líneas 1 a 3). La iteración comienza a buscar alguna mejora para un vértice del conjunto de dominancia (línea 4). A esta nueva solución se le aplica el proceso de agujereado (línea 5) que se detallará más adelante en el Algoritmo 3. Posteriormente, se reconstruye la solución utilizando el mismo criterio voraz descrito en el constructivo RGA (Sección 2.1), utilizando un valor de $\alpha$ completamente voraz, que se ha determinado experimentalmente (línea 6). Finalmente, se eliminan aquellos nodos redundantes (línea 7) al igual que en el Algoritmo 1 y, si la solución obtenida es mejor que la anterior (línea 8), se actualiza la mejor solución (línea 9), comenzando la búsqueda de nuevo (línea 11). En caso contrario, se continúa buscando una mejora sobre el siguiente nodo del conjunto $D$.

---

**Algoritmo 2** *Mejorar* $(D, \delta)$

---
1: $mejora \leftarrow$ VERDADERO
2: **while** $mejora$ **do**
3:    $mejora \leftarrow$ FALSO
4:    **for all** $v \in D$ **do**
5:       $D' \leftarrow Agujerear(D, v, \delta)$
6:       $D'' \leftarrow Reconstruir(D')$
7:       $D''' \leftarrow EliminarRedundantes(D'')$
8:       **if** $|D'''| < |D|$ **then**
9:          $D \leftarrow D'''$
10:         $mejora \leftarrow$ VERDADERO
11:         **go to** 3
12:       **else**
13:          $D' \leftarrow D$
14:       **end if**
15:    **end for**
16: **end while**
17: **return** $D$

---

En el Algoritmo 3 se presenta el pseudocódigo del proceso de agujereado, el cual pretende realizar agujeros en un conjunto de dominancia $D'$ mediante la eliminación de varios nodos partiendo de un nodo inicial $v$, evitando eliminar aquellos fijados previamente por las reglas de reducción. Durante este proceso se eliminarán todos los nodos que se encuentran a $\delta$ niveles de profundidad del nodo $v$ en el grafo, siendo la vecindad con $\delta = 1$ el conjunto $\{v\}$. En este procedimiento, si no se ha alcanzado el último nivel de expansión (línea 1), se elimina el vértice $v$ (línea 2) y para cada uno de los vértices adyacentes (línea 3) se llama al algoritmo con un nivel de expansión inferior (línea 4), pues el nivel que correspondería al vértice $v$ ha sido realizado en la línea 2. Finalmente, se devuelve el conjunto de dominancia $D$ generado (línea 7).

---

**Algoritmo 3** *Agujerear* $(D, v, \delta)$

---
1: **if** $\delta > 0$ **then**
2:    $D \leftarrow D \setminus \{v\}$
3:    **for all** $u \in N(v)$ **do**
4:       $Agujerear(D, u, \delta - 1)$
5:    **end for**
6: **end if**
7: **return** $D$

---

Partiendo de las soluciones incompletas, se reconstruirá el conjunto dominante, añadiendo los nodos necesarios para que éste sea factible utilizando la heurística voraz basada en el grado descrita en la Sección 2.1. En este caso, se mantendrá únicamente el criterio voraz debido al coste computacional en caso de utilizar una variante semialeatoria.

Para ilustrar de forma gráfica el método, la Figura 1 muestra ejemplos de la fase de mejora del proceso GRASP implementado. En estas figuras se observa una red social representada como un grafo con 9 nodos y 12 aristas en diferentes momentos de la fase de agujerado. Estos nodos se encuentran

identificados con diferentes colores donde cada color representa un estado diferente. Los verdes son los incluidos en el conjunto solución, los negros son los fijados por la regla de reducción de los nodos hoja y los grises por la regla de reducción relacionada con los triángulos. Por otra parte, los nodos blancos no se encuentran en la solución y los rojos quedarían excluidos por la regla de reducción del triángulo [21]. Finalmente, en naranja se indica el nodo inicial de la propagación de la búsqueda local por agujeros y en amarillo aquellos afectados por el proceso de agujereado.

Figura 1: Instancia compuesta de 9 nodos y 12 relaciones, antes 1a, durante 1b y después 1c del proceso de búsqueda local por agujeros.

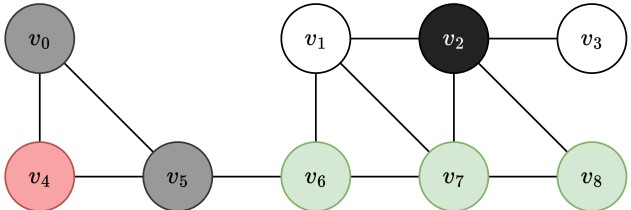

(a) Ejemplo de solución dada por una construcción inicial.

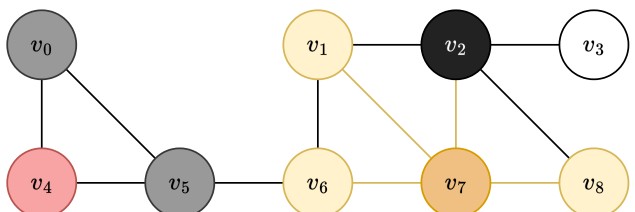

(b) Propagación de la búsqueda local por agujeros con $\delta = 2$.

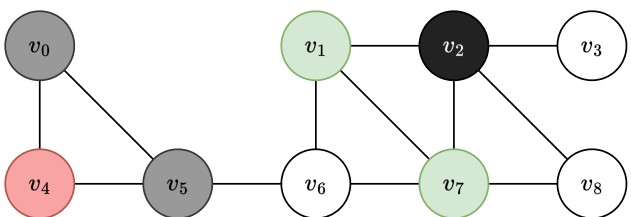

(c) Solución MPIDS tras la búsqueda local por agujeros.

La solución inicial propuesta por la fase constructiva de la Figura 1a cuenta con una función objetivo $|D| = |\{v_0, v_2, v_5, v_6, v_7, v_8\}| = 6$. Aplicando la búsqueda local por agujeros sobre el nodo $v_7$, se ven afectados los nodos $\{v_1, v_6, v_7, v_8\}$, que se eliminan del conjunto solución (Figura 1b). Cabe destacar que el nodo $v_2$ también debería verse afectado, pero no lo hace al estar marcado como fijo por la reducción de los nodos hoja. Tras la posterior reconstrucción voraz del conjunto solución (Figura 1c), se obtiene una nueva solución tal que $|D| = |\{v_0, v_1, v_2, v_5, v_7\}| = 5$, la cual sería una solución de cardinalidad mínima para el problema MPIDS.

## 3. Experimentos

En esta sección se analizan los resultados obtenidos en las diversas experimentaciones realizadas. Se ha ejecutado el algoritmo diseñado sobre 196 instancias recopiladas de la literatura, obtenidas de repositorios públicos como *Stanford Network Analysis Project* (SNAP) [10] o *Network Respository* [20].

Tabla 1: Características de las instancias

| Medida | Total | Promedio | Mín. | Máx. |
|---|---|---|---|---|
| Tamaño (B) | 33 304 329 818 | 170 791 434 | 492 | 4 311 190 746 |
| # Nodos | 415 204 269 | 2 129 253 | 34 | 59 216 211 |
| # Aristas | 2 100 272 636 | 10 770 628 | 78 | 261 321 071 |

La Tabla 1 recoge las características principales de estas instancias. Se incluye un resumen estadístico de tres medidas clave: tamaño en bytes, número de nodos y número de aristas, junto con sus valores totales, promedio, mínimo y máximo. En total, se consideran más de 415 millones de nodos, donde algunas redes son extremadamente pequeñas (34 nodos), mientras que la más grande supera los 59 millones. Por otra parte, se registran más de dos mil cien millones de aristas, donde red más pequeña contiene 78 conexiones, mientras que la más grande supera los 261 millones.

La experimentación diseñada comprende dos fases y se ha realizado en una máquina virtual de un servidor con un procesador AMD EPYC 7282 (2.8 GHz), 128 GB de memoria RAM, Ubuntu Server 20.04 y Java 21. La primera fase incluye un experimento preliminar realizado sobre un subconjunto diverso de instancias (sección 3.1), cuya finalidad es definir los mejores valores de parámetros para el algoritmo. Para evitar sobreajustes en la configuración de los parámetros [7], ese subconjunto se obtiene de la selección de un 25 % de las instancias disponibles. La segunda fase consiste en una experimentación final para validar la mejor configuración frente al estado del arte (sección 3.2).

Todos los experimentos se han evaluado utilizando las mismas métricas, entre las que se incluye: *Promedio*, que representa la media del valor de la función objetivo de todas las instancias para un algoritmo; *Tiempo (s)*, que señala el tiempo medio de ejecución en segundos; *Desv. ( %)*, que muestra la media de la desviación porcentual de la función objetivo frente al mejor valor conocido del experimento para cada instancia; y *# Mejores*, que indica el número de mejores soluciones encontradas en el experimento. Asimismo, se ha indicado en negrita en cada una de las tablas de esta sección el mejor de los valores en cada una de estas métricas en cada experimento.

Cabe destacar que, en la literatura, establecen un criterio de parada de una hora para obtener una solución para algunas de las instancias y de 1000 segundos para otras. Por este motivo, en la experimentación realizada se ha limitado el tiempo de ejecución de los algoritmos propuestos a una hora.

### 3.1. Experimento preliminar

La Tabla 2 permite analizar la contribución de las diferentes aproximaciones estudiadas: i) una única construcción aleatoria de la solución; ii) varias ejecuciones aleatorias limitando el tiempo de ejecución a una hora; iii) una única construcción voraz; y iv) una construcción voraz con eliminación de nodos redundantes.

Tabla 2: Comparativa de construcciones aleatorias y voraces.

| Algoritmo | Promedio | Tiempo (s) | Desv. ( %) | # Mejores |
|---|---|---|---|---|
| *Aleatorio* | 776363.86 | 3.13 | 23.23 | 0 |
| *Aleatorio (3600 s)* | 776157.63 | 3086.22 | 19.02 | 3 |
| *Voraz* | 650048.98 | **2.74** | 20.44 | 1 |
| *Voraz\** | **621659.84** | 2.87 | **0.42** | **46** |

Se puede observar como el criterio voraz utilizado es adecuado para este problema, puesto que es capaz de obtener uno de los menores valores promedio de la función objetivo, una desviación baja, en un tiempo muy reducido, en comparación con el algoritmo aleatorio que emplea 3600 segundos. Destaca especialmente la desviación de esta ejecución, pues aun obteniendo una reducción de más de

100000 nodos en el promedio, obtiene un valor ligeramente superior que el algoritmo *Aleatorio(3600 s)*. Esto se debe a la gran diferencia en los tamaños de las instancias, pues una diferencia de 3 o 5 nodos muestra entre un 20 % y 30 % de desviación respecto al mejor, mientras que para alcanzar esa desviación en instancias grandes debe haber una diferencia de 0.5 millones de nodos. Como "Voraz" tiene mejores resultados en instancias grandes y peores en las pequeñas, se encuentra penalizado con una mayor desviación aún habiendo reducido el promedio.

Por otra parte, el algoritmo *Voraz\** que cuenta con la eliminación de nodos redundantes muestra aún mejores resultados que el algoritmo voraz. Con un ligero aumento del tiempo de ejecución, obtiene el mejor valor en 46 del total de 49 instancias, dejando atrás al resto de propuestas en cuanto a valor de la función objetivo y desviación. Esto demuestra que el proceso de eliminación de nodos redundantes contribuye de forma positiva en la propuesta. En consecuencia, el método *Voraz\** es el elegido para validar el tamaño $\delta$ de agujeros.

Para obtener la mejor configuración de la búsqueda local por agujeros, se han realizado dos ejecuciones con diferentes valores del parámetro $\delta$, recogidas en la Tabla 3. Estos valores han sido 2 y 3, ya que $\delta = 1$ únicamente se eliminaría a sí mismo y valores de $\delta > 3$ podría llegar a eliminar la solución prácticamente al completo en instancias pequeñas o con un grado de conectividad alto.

Tabla 3: Comparativa de diferentes valores para el parámetro $\delta$ en la fase de agujereado.

| $\delta$ | Promedio | Tiempo (s) | Desv. ( %) | # Mejores |
|---|---|---|---|---|
| 1 | 621659.84 | **2.87** | 4.25 | 3 |
| 2 | 565363.71 | 252.13 | **0.13** | **40** |
| 3 | **565325.10** | 675.54 | 0.54 | 13 |

Con esta experimentación, podemos observar que, aunque con un valor $\delta = 3$ obtengamos una ligera mejora en cuanto al promedio del valor de la función objetivo, los resultados con $\delta = 2$ tienen una menor desviación y un mayor número de mejores soluciones con un tiempo notablemente inferior. Esto resulta interesante, ya que para instancias grandes, cuanto menor sea el tiempo que requiera la búsqueda local, mayor número de iteraciones de la metaheurística GRASP se podrán realizar, permitiendo así encontrar mejores soluciones.

Finalmente, para encontrar la mejor configuración del parámetro $\alpha$ en el constructivo RGA se han realizado cinco ejecuciones GRASP con $\delta = 2$ y $\alpha = \{0.00, 0.25, 0.50, 0.75, 1.00\}$ además de una sexta ejecución $\alpha = RND$ donde se genera un valor aleatorio uniforme en $\alpha = [0.00, 1.00]$ para cada nueva construcción. El tiempo de ejecución considerado sigue siendo de una hora para todas las configuraciones salvo para la de $\alpha = 0.00$ que, al ser un constructivo voraz, siempre parte de la misma solución inicial. Los resultados obtenidos se muestran en la Tabla 4.

Tabla 4: Comparativa para obtener la mejor configuración del parámetro $\alpha$ en la fase de construcción.

| $\alpha$ | Promedio | Tiempo (s) | Desv. ( %) | # Mejores |
|---|---|---|---|---|
| *RND* | 593147.63 | 3600.23 | 0.44 | 5 |
| 0.00 | **565363.71** | **252.13** | 1.30 | 6 |
| 0.25 | 615990.47 | 3600.16 | 0.51 | 12 |
| 0.50 | 593060.12 | 3600.17 | 0.35 | 15 |
| 0.75 | 579615.39 | 3600.22 | **0.25** | 14 |
| 1.00 | 643084.12 | 3600.26 | 0.63 | **24** |

Como se puede observar, el valor $\alpha = 0.00$ es el que ha obtenido mejor promedio, con un tiempo menor, ya que su criterio generativo es totalmente voraz. Por otra parte, la ejecución con $\alpha = 1.00$ ha obtenido una mayor calidad en cuanto a mejores resultados por número de instancias, pero obteniendo un valor promedio alto en comparación al resto de resultados, lo cual sugiere un rendimiento bueno en instancias pequeñas, pero una debilidad con instancias grandes.

Finalmente, se ha decidido utilizar la configuración $\alpha = 0.75$ ya que además de obtener una calidad competente con los dos anteriores valores de $\alpha$ mencionados, la mejor desviación con un 0.25 % y el segundo mejor valor en términos de promedio de la función objetivo, aporta una gran variabilidad en la fase de construcción.

### 3.2. Experimento final

Una vez obtenida la mejor configuración de la implementación realizada, se ha realizado una experimentación final con un constructivo aleatoriamente voraz con $\alpha = 0.75$, una eliminación de nodos redundantes y un proceso de mejora de búsqueda local por agujeros con dos niveles de expansión $\delta = 2$. Para poder realizar una comparativa con los mejores resultados del estado del arte, obtenidos por el algoritmo FastPIDS diseñado por Sun et al. [21], se ha limitado el tiempo de ejecución a una hora. Además, hemos seguido la misma metodología que los autores previos, realizando diez ejecuciones con diferentes semillas aleatorias. La Tabla 5 recoge los resultados promedios de todas las ejecuciones realizadas.

Tabla 5: Comparativa del algoritmo FastPIDS del estado del arte y nuestra propuesta con los valores promedio de las diez ejecuciones.

| Algoritmo | Promedio | Tiempo (s) | Desv. ( %) | # Mejores |
|---|---|---|---|---|
| FastPIDS | **1023870.82** | **3600.00** | **0.00** | **195** |
| GRASP | 1069583.08 | 3629.33 | 5.54 | 4 |

La evaluación realizada refleja que el algoritmo previo obtiene mejores resultados en todas las medidas consideradas, con un menor valor promedio y alcanzando los mejores resultados sobre el total de 195 instancias evaluadas. Sin embargo, la solución metaheurística propuesta obtiene unos resultados razonables, con una desviación cercana al $5.54$ %. Los resultados obtenidos, aun no superando el estado del arte, muestran como un diseño metaheurístico basado en la combinación de varios métodos sencillos puede mostrar un rendimiento competitivo. La siguiente sección muestra algunas conclusiones y trabajos futuros descritos que plantean ideas prometedoras para seguir mejorando estos enfoques.

## 4. Conclusión

En este trabajo se ha propuesto una aproximación GRASP para el problema de dominancia MPIDS en redes sociales. Se ha diseñado una fase de construcción eficiente que aporta una gran variabilidad en proceso de obtención de soluciones iniciales sobre las que, en una posterior fase de mejora, se consigue reducir considerablemente el tamaño del conjunto de dominancia. Por otra parte, la búsqueda local por agujeros propuesta genera óptimos locales a partir de las soluciones iniciales dadas en un tiempo de cómputo reducido. De esta manera, GRASP ofrece resultados competitivos respecto a FastPIDS. Algunos trabajos futuros derivados de la investigación son explorar nuevos algoritmos metaheurísticos que puedan mejorar la diversidad de las soluciones obtenidas, además de realizar un estudio de las características de las mejores soluciones para aprovecharlas, creando diversos movimientos o algoritmos constructivos inteligentes. Por último, nos planteamos realizar una experimentación factorial mediante herramientas como iRace [13] para obtener la mejor configuración del algoritmo, seleccionando las instancias preliminares de forma más cuidadosa [15].

## Agradecimientos y declaración de financiación

Los autores agradecen el apoyo del "Ministerio de Ciencia, Innovación y Universidades (MCIN/AEI/10.13039/501100011033/FEDER, UE) bajo la subvención ref. PID2021-126605NB-I00 y PID2021-125709OA-C22, y del "Ministerio para la Transformación Digital y de la Función Pública" mediante Concesión TSI-100930-2023-3.

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
