# OpenReview forum: "Resolución del problema de conjunto dominante de influencia positiva mínima en redes sociales mediante metaheurísticas"
_MAEB/2025/Congreso — MAEB 2025_

### Official Review · Reviewer_HQhD · 2025-03-11
**Resolución del problema de conjunto dominante de influencia positiva mínima en redes sociales mediante metaheurísticas**

**Rating:** 5
**Confidence:** 2

**Review:**

Este trabajo presenta una aproximación GRASP para el problema de dominancia MPIDS (Minimum Positive Influence Dominating Set) en redes sociales. Se desarrolla una fase de construcción que proporciona variabilidad en las soluciones iniciales, seguida de una fase de mejora que reduce significativamente el tamaño del conjunto de dominancia. La búsqueda local por agujeros optimiza las soluciones, logrando resultados competitivos frente a FastPIDS, que actualmente el de mejor rendimiento según la literatura.

El artículo está bien escrito y resulta interesante, lo que facilita su lectura y comprensión. Presenta un algoritmo que es equiparable a los mejores existentes en términos de rendimiento y eficiencia.

---

### Official Review · Reviewer_QtSb · 2025-03-15
**Aplicación de GRASP al problema MPIDS**

**Rating:** 4
**Confidence:** 4

**Review:**

El trabajo se encuentra en el ámbito de estudio de la influencia en redes sociales, concretamente aborda la maximización de influencia social.
El problema MPIDS trata de buscar un conjunto mínimo de usuarios que permita influir en toda la red.

Los autores utilizan FastPIDS como algoritmo base para la comparación de resultados.

El artículo está bien escrito, y detalla una metodología razonable, con una experimentación que muestra resultados prometedores.  Los autores reconocen que el método propuesto aún no supera el estado del arte aunque se queda cerca, y muestra el camino hacia el trabajo futuro de mejora.

Por todo lo anterior, y por la naturaleza del congreso MAEB, se considera un trabajo aceptable para el mismo.

---

### Official Review · Reviewer_4m36 · 2025-03-17
**Buen paper**

**Rating:** 5
**Confidence:** 2

**Review:**

Este paper propoen un metodo heuristico para el problema MPIDS, que consiste en encontrar el subconjunto mas pequeño posible que influya en toda la red social (el grafo completo). Para un nodo es influenciado cuando al menos la mitad de los vecinos deben estar en el subconjunto de dominancia (ser parte de la solucion). El metodo descrito en este paper propoen un constructivo voraz + eliminacion de nodos redundantes + busqueda local basada en eliminacion de soluciones vecinas. El metodo no consigue mejorar el estado del arte.


Es un buen paper con malos resultados, aun asi merece ser publicado. Propongo estas mejoras/cuestiones a plantearse.

- Cual es la contribucion de cada "componente"? Estaria bien comparar el metodo cuando se considera sin cada una de las partes: sin construccion voraz, sin eliminacion de nodos redundantes o sin busqueda local. Asi podriamos saber como de importante es cada una de estas componentes.

- La intencion de la Figura 1 es buena, pero me cuesta entenderlo. Creo que se ponene demasiadas cosas en cada figura. Tal vez se podrian poner 8 subfiguras en vez de 3 y poner en cada una solamente un concepto, para que fuera mas facil de entender.

- No seria posible combinar el metodo del estado del arte con los metodos de eliminacion de nodos redundantes + busqueda local que proponeis? Tal vez esto podria mejorar los resultados.

---

### Decision · Program_Chairs · 2025-03-20

Accept